# Radical-Generating Activity, Phagocytosis, and Mechanical Properties of Four Phenotypes of Human Macrophages

**DOI:** 10.3390/ijms25031860

**Published:** 2024-02-03

**Authors:** Shakir K. Suleimanov, Yuri M. Efremov, Timofey O. Klyucherev, Emin L. Salimov, Aligeydar A. Ragimov, Peter S. Timashev, Irina I. Vlasova

**Affiliations:** 1Institute for Regenerative Medicine, I. M. Sechenov First Moscow State Medical University, 119991 Moscow, Russia; suleymanovef@gmail.com (S.K.S.); efremov_yu_m@staff.sechenov.ru (Y.M.E.); klyucherev_t_o@staff.sechenov.ru (T.O.K.); timashev_p_s@staff.sechenov.ru (P.S.T.); 2Laboratory of Clinical Smart Nanotechnologies, I. M. Sechenov First Moscow State Medical University, 119991 Moscow, Russia; 3Laboratory Blood Transfusion Complex, I. M. Sechenov First Moscow State Medical University, 119991 Moscow, Russia; salimov_e_l@staff.sechenov.ru (E.L.S.); ragimov_a_a@staff.sechenov.ru (A.A.R.); 4World-Class Research Center “Digital Biodesign and Personalized Healthcare”, I. M. Sechenov First Moscow State Medical University, 119991 Moscow, Russia

**Keywords:** myeloid-derived macrophages, reactive oxygen species, redox activity, phagocytosis, Young’s modulus

## Abstract

Macrophages are the major players and orchestrators of inflammatory response. Expressed proteins and secreted cytokines have been well studied for two polar macrophage phenotypes—pro-inflammatory M1 and anti-inflammatory regenerative M2, but little is known about how the polarization modulates macrophage functions. In this study, we used biochemical and biophysical methods to compare the functional activity and mechanical properties of activated human macrophages differentiated from monocyte with GM-CSF (M0_GM) and M-CSF (M0_M) and polarized into M1 and M2 phenotypes, respectively. Unlike GM-CSF, which generates dormant cells with low activity, M-CSF confers functional activity on macrophages. M0_M and M2 macrophages had very similar functional characteristics—high reactive oxygen species (ROS) production level, and higher phagocytosis and survival compared to M1, while M1 macrophages showed the highest radical-generating activity but the lowest phagocytosis and survival among all phenotypes. All phenotypes decreased their height upon activation, but only M1 and M2 cells increased in stiffness, which can indicate a decrease in the migration ability of these cells and changes in their interactions with other cells. Our results demonstrated that while mechanical properties differ between M0 and polarized cells, all four phenotypes of monocyte-derived macrophages differ in their functional activities, namely in cytokine secretion, ROS production, and phagocytosis. Within the broad continuum of human macrophages obtained in experimental models and existing in vivo, there is a diversity of phenotypes with varying combinations of both markers and functional activities.

## 1. Introduction

Macrophages are innate immune cells that play a crucial role in all phases of the immune response. They defend against pathogens in acute-phase response, participate in tissue repair, and orchestrate the entire process of regeneration. 

Upon development of inflammation, monocytes migrate to an inflammatory site, where they undergo differentiation into macrophages as a result of exposure to tissue cytokines released by surrounding cells such as fibroblasts, endothelial cells, stromal cells, etc. [1,2,3]. Based on the signals received from the microenvironment, macrophages change the expression of genes, morphology and their specific functions in the processes called differentiation and polarization [4]. This process is mainly mediated by growth factors like granulocyte–macrophage colony-stimulating factor (GM-CSF) and macrophage colony-stimulating factor (M-CSF) [5]. Then, under the influence of a variety of biologically active compounds like cytokines and lipopolysaccharides (LPS), macrophages are polarized to various phenotypes. The traditional M1/M2 paradigm distinguishes two main macrophage subtypes depending on their phenotypic features, cytokine secretion, and transcriptional profiles: M1 (classically activated) and M2 (alternatively activated) macrophages [6,7,8]. LPS in combination with interferon-γ (IFN-γ) are the major inducers of macrophage polarization into M1 phenotype [3]. M1 macrophages participate in antigen presentation and produce reactive oxygen species (ROS), inflammatory factors (including interleukine-1β (IL-1β), IL-6, IL-12, and IL-23), and tumor necrosis factor TNF-α [9,10]. To differentiate macrophages to the precursor of M2 phenotype, the monocytes should be pre-incubated with M-CSF [11]. Then, polarization into the M2 state is accomplished by a number of inducers. Depending on the inducer, one of the four subtypes of M2 macrophages can be obtained. Incubation of macrophages with IL-4 results in the formation of M2a phenotype [12]. M2 macrophages demonstrate high phenotypic plasticity and anti-inflammatory and immunoregulatory activity; they contribute to the resolution of inflammation and regeneration due to the production of growth factors and anti-inflammatory cytokines IL-10, CCL18, and IL-1Ra. M2 macrophages express the mannose receptor CD206, known as a phagocytic receptor for pathogens, and CD163, a receptor for haptoglobin–hemoglobin complexes [13].

The polarization of M1 and M2 macrophages is a highly controlled process, and an imbalance of macrophage polarization is often associated with various diseases or inflammatory conditions, such as rheumatoid arthritis, atherosclerosis, cancer, and diabetes [2,12,13,14]. Balance between pro-inflammatory and anti-inflammatory macrophages is essential for the activated-to healing transition. Macrophage plasticity is considered a therapeutic target in different pathologies [15,16]. 

The radical-generating ability of macrophages is one of the most important activities of these cells both under normal physiological conditions and during inflammation. Under normal conditions, ROS play a crucial role in signal transduction, differentiation, gene expression, and apoptosis. NADPH oxidase (mainly NOX2) is the major source of ROS in macrophages [17]. A growing body of evidence points at mitochondria as essential sites of ROS formation, mainly due to electron leakage of the respiratory chain or to enzymes, such as monoamine oxidases [18]. Under physiological conditions, ROS production is important for the differentiation of human monocytes into M2 macrophages [19]. The levels of ROS and the production of extracellular H_2_O_2_ are lower in M2 macrophages compared to M1 cells due to a decreased expression of redox-active enzymes, most particularly NOX2 and NOX5 as well as higher levels of antioxidant enzymes [20]. 

In the beginning of inflammation, due to a high concentration gradient of GM-CSF and proinflammatory cytokines such as IFN-γ at the site of injury, a massive influx of monocytes from blood vessels gives rise to a population of M1 macrophages. M1 cells are activated at the site of inflammation and produce ROS and proinflammatory cytokines. The respiratory burst of activated macrophages is associated predominantly with NOX2, which produces a high level of the superoxide anion radical (O_2_^•−^) [21]. Spontaneously or under the catalytic activity of superoxide dismutase (SOD), O_2_^•−^ tunes into hydrogen peroxide. H_2_O_2_ reacts with iron in low-molecular complexes or in the active sites of enzymes with the production of ROS, thus initiating oxidative stress outside and inside the cells [22,23]. Upon development of an inflammatory response, the macrophage polarization is shifted into M2 phenotype. M2 macrophages exhibit multiple functions, including the secretion of anti-inflammatory cytokines, phagocytosis of apoptotic cells, and ROS production [3,9]. It is generally accepted that in conditions of acute inflammation, M1 macrophages have the highest radical-generating activity, creating a pro-oxidative environment and contributing to the killing and elimination of bacteria, whereas M2 macrophages remove damaged cells and matrix and are involved in tissue repair. 

ROS production is closely related to another important function of macrophages—phagocytosis. After assembly on the membrane, NOX2 synthesizes the superoxide anion radicals inside the phagosome, which, along with proteolytic enzymes, ensures the destruction of a captured material [21]. Phagocytic activity is a characteristic of both M1 and M2 macrophages, which was shown in RAW 264.7 macrophages [24]. However, some studies demonstrated a decrease in the phagocytic activity of macrophages after polarization towards the M2 phenotype. IL-4-induced macrophage polarization towards an alternatively activated phenotype was demonstrated to cause a significant decrease in phagocytosis of N. meningitidis, but at the same time enhances the production of cathepsins and increases the acidity and proteolytic activity of phagosomes [25,26]. On the contrary, studies of human macrophages demonstrate that the level of E. coli phagocytosis in M2 macrophages is higher compared to M1 macrophages [27]. Phagocytic activity of macrophages may depend on the specific pathogen used in a study, which hinders making general conclusions on the results.

The redox activity of macrophages has been studied mainly using mouse macrophages, while human macrophages differ from mouse cells. For example, mouse M1 macrophages produce high amounts of NO, key storage (GSPT1), and transport (MRP1) proteins are engaged to protect professional killer cells from this cytotoxic agent [28]. On the contrary, despite the expression of iNOS, human macrophages do not produce a significant amount of nitric oxide, which indicates low activity of the enzyme [29,30,31]. At the same time, high activity of iNOS was reported under some pathological conditions [32]. In addition, arginase-1, the main marker of mouse M2 macrophages, is weakly expressed in human macrophages [7]. The clear dichotomy of polarization demonstrated for mouse macrophages is not as obvious in humans, and the separating line between M1-like and M2-like macrophages is rather represented by a continuum of phenotypes [9].

The study of the redox activity of human macrophages is important to understand the roles of M1 and M2 macrophages in the immune system. Using luminol-dependent chemiluminescence (CL), Lewis et al. showed similar radical-generating activity and found no differences in NOX2 expression in M1 and M2 human macrophage phenotypes either for THP-1-derived macrophages or monocyte-derived macrophages (MDM) [33]. The authors used high concentrations of phorbol 12,13-dibutyrate (10 μM) to activate macrophages. Nassif et al. demonstrated that the hypoglycemic agent metformin inhibits LPS-induced radical-generating M2 activity in human macrophages via the activation of adenosine monophosphate protein kinase [34].

Macrophage polarization and activation evokes changes in cytoskeleton structure and membrane composition, what results in changes of cell morphology and membrane stiffness. AFM is a powerful tool to study the cellular morphology and bio-mechanics of immune cells [35]. Activation of RAW264.7 macrophages by LPS (100 ng/mL) caused changes in cell morphology and membrane ultrastructure and increased the Young’s modulus of cells, which was attributed to the redistribution of intracellular F-actin structures upon LPS treatment [36]. On the contrary, using colloidal force microscopy Leporatty et al. found that the stiffness of human macrophages decreased after cell stimulation by 10 µg/mL LPS [37]. The mechanical properties of macrophages are important in their interaction with the cell matrix and other cells involved in tissue regeneration and affect cell proliferation and migration [38]. 

In this study, we compare the functional activity and mechanical properties of four phenotypes of human macrophages with different phenotypic and secretory features. Monocytes were treated with GM-CSF (M0_GM) or M-CSF (M0_M) and then activated into M1 or M2 states, respectively. To simulate the conditions of the inflammatory site, we tested the functional activity and mechanics of macrophages in the presence of radical-generating inducers (low molecular phorbol-12-myristate-13-acetate (PMA, MW = 617 g/mol) and particulate opsonized zymosan (OZ)) and bacteria. Four phenotypes of macrophages differed in cytokine secretion, ROS generation, and ability to phagocytosis. For all macrophages studied, a similar decrease in viscoelastic parameter and height was observed upon activation, that evidenced cytoskeleton rearrangement, but only M1 and—to a lesser extent—M2 cells increased their rigidity. Changes in mechanical properties did not correlate with macrophage functional activity.

## 2. Results

### 2.1. Monocyte Differentiation and Macrophage Polarization

Macrophages were characterized by the expression of surface proteins, which serve as phenotype markers: CD86 indicated M1 phenotype, and CD206 indicated M2 phenotype. Macrophage functional response was determined by the secretion of cytokines—TNF-α for M1 and IL-10 for M2. 

Treatment of monocytes with CSFs stimulates their differentiation into M0-macrophages. Incubation of macrophages with GM-CSF led to increased expression of CD86 and a shift of the anti-CD206-PE-Cy7 median fluorescence intensity (MFI) compared to cells treated with M-CSF. At the same time, low secretion of TNF-α and no secretion of IL-10 was observed for M0_GM cells, while cells incubated with M-CSF secreted a high amount of IL-10 (Figure 1). Further polarization of M0_GM macrophages into the M1 state (+LPS+IFN-γ) led to an increase in surface expression of CD86, an amplification of TNF-α production by about a hundredfold and secretion of a small amount of IL-10. IL-4-treated M2 macrophages were characterized by the highest expression of CD206 compared to other phenotypes and showed a slight increase in IL-10 production compared to their precursor M0_M macrophages (Figure 1). Another surface protein CD163 was not a good marker for M2 obtained after MDM treatment with IL-4, its expression was similar in four phenotypes of macrophages (Appendix A). CD163 was shown to be highly expressed on IL-10-treated MDM [13,39]. IL-1Ra can be considered as another marker of the M2a phenotype (Appendix A) [13]. Cellular morphology serves as an additional polarization marker. M2 macrophages predominantly exhibit an oblong shape, whereas M1 macrophages typically have a more rounded, spherical shape. (Appendix A) [20].

### 2.2. Effect of Growth Factors and Polarization Inducers on Macrophage Survival

In our experiments, the monocytes were seeded at the same number per well of the plate. However, by the sixth day of differentiation with CSF, a difference in cell numbers between the M0_GM and M0_M groups was already observed. The differences between cells increased after macrophage polarization into M1 and M2 phenotypes. To assess differences, the concentration of double-stranded DNA was measured using PicoGreen dye since the amount of DNA correlates with the number of cells. The relative difference between groups in each individual experiment, rather than the absolute values of DNA concentrations, was important. Data of four independent experiments are presented in Figure 2 as a heatmap reflecting levels of DNA (ng/mL) lower (blue shades) and higher (red shades) than expected values for each sample. The results showed that the group of macrophages incubated with M-CSF had a higher number of viable cells compared to macrophages incubated with GM-CSF. Moreover, the number of living cells decreased after M0_GM polarization to M1 state. The addition of IL-4 to M-CSF treated macrophages also resulted in a slight decrease in the number of viable cells compared to M0_M.

The results of DNA measurements were used to normalize the measured ROS parameters to the number of cells in a sample in order to compare the radical-generating activity of different phenotypes of macrophages.

### 2.3. ROS Generation by Activated Macrophages

Along with the secretion of cytokines, an important function of macrophages is ROS production. To simulate the conditions of the inflammatory site, macrophages were activated. We compared the radical-generating activity of different macrophage phenotypes after cell activation by substances with different mechanisms of action—opsonized zymosan and phorbol-12-myristate-13-acetate. Simultaneously with superoxide generation, activated macrophages release SOD3 from intracellular compartments, which catalyzes the dismutation of O_2_^•−^ into hydrogen peroxide [40]. To measure H_2_O_2_, we used horseradish peroxidase (HRP) and two peroxidase substrates—luminol and 10-acetyl-3,7-dihydroxyphenoxazine (Amplex Red).

#### 2.3.1. Luminol-Dependent Chemiluminescence

Luminol is able to penetrate into cells. Therefore, using luminol, we measured extra- and intracellular ROS [41]. After activation with OZ, macrophages cultured in the presence of GM-CSF showed a very low level of luminol oxidation, while macrophages polarized to M1 state produced high levels of ROS: the luminol oxidation increased within 10–20 min, reached a maximum, and slowly decreased (Figure 3A). In the presence of PMA, M0_GM cells also demonstrated a slight increase in CL (Figure 3B). For both cellular activators, CL (area under CL curve) of M0_GM cells was approximately an order of magnitude lower than that for macrophages polarized in M1 phenotype (Figure 3C).

The addition of any of the activators to both types of macrophages differentiated in the presence of M-CSF caused CL comparable to the M1 level. Notably, M2 macrophages reached CL maximum significantly later than their precursor M0_M macrophages. Moreover, both M2 and M0_M macrophages activated with PMA reached CL intensity peak faster than those activated with OZ (Figure 3D,E and Appendix A). Significant differences in the intensity of the luminescence peaks of M0_M and M2 macrophages for both activators were not found (Figure 3F).

Additionally, experiments were performed with macrophages detached by accutase; CL was measured in a cell suspension. The ratio of activities of different macrophage phenotypes in the suspensions was consistent with the results from experiments with adherent cells (Appendix A). We compared pro-inflammatory properties of detached M1 macrophages derived from MDM and THP-1 macrophages. Polarization of THP-1 macrophages into M1 state was evidenced by secretion of pro-inflammatory cytokines TNF-a and IL-1β, surface expression of CD86 and CD206 did not differ between groups (Appendix A). Both the ROS-generating activity and the cytokine secretion by M1 macrophages differentiated from leukemia cells were approximately 10 times lower than the activity of professional macrophages in M1 state (Appendix A). [33]. Taking into account the difference between the level of secreted cytokines [42,43], the direct comparison of MDM to THP-1 macrophages is not relevant.

#### 2.3.2. Oxidation of Amplex Red by Horseradish Peroxidase in the Presence of Activated Macrophages

Contrary to luminol, Amplex Red cannot pass through the cell membrane. Using this reagent, we measured H_2_O_2_ outside the cells and in phagosomes. Activation of M0_GM by OZ resulted in a weak increase in Amplex Red oxidation and, accordingly, in an enhancement of fluorescence (Figure 4A). Notably, the response of M0_GM cells to the activator differed by an order of magnitude in different experiments. However, it was significantly weaker than the response of M1 cells (Figure 4C). Activation of M1 cells with OZ caused time-dependent amplification of fluorescence with a maximum at about 20 min. Similar to OZ, the fluorescence in the solution of PMA-activated M1 macrophages was significantly higher than that in the solution of precursor M0_GM (Figure 4B). For both activators, the wide scatter in the generated radical data of M0_GM group was observed (Figure 4C).

Both activators induced ROS generation by M0_M and M2 macrophages, increasing the fluorescence of resorufin in the medium (Figure 4D,E). Radical-generating activity of both types of cells treated with M-CSF was comparable to that of M1 phenotype, but the kinetics of accumulation of the fluorescent product was different—the maximum of fluorescence was reached no earlier than 30 min after the addition of any activator. Moreover, in contrast to CL measurements, the level of ROS production of M-CSF treated cells activated by PMA was statistically significantly lower if cells were polarized into the M2 phenotype (Figure 4F).

### 2.4. Comparison of ROS-Generating Ability of M1 and M2 Macrophages

ROS-generating activities (area under the curves) of M1 and M2 macrophage phenotypes are shown in Figure 5. Comparison of the level of ROS generation by macrophages activated by different inducers and measured by different methods revealed general trends: (1) M1 macrophages demonstrated the highest radical-generating activity; (2) a significant difference in the ROS-generated activity of macrophages M1 and M2 was observed upon activation by PMA, but not by the particulate agonist OZ (compare Figure 5A,C); (3) polarization of M0_GM macrophages into the M1 state amplified the ROS activity of cells, while polarization of M0_M macrophages into the M2 state does not change the level of total ROS production (CL) but slightly decreased the generation of H_2_O_2_ outside the cells (Amplex Red) (Figure 5B,D).

### 2.5. Macrophage Phagocytosis

We employed flow cytometry and fluorescent microscopy to compare phagocytosis of inactivated FITC-labeled E. coli bioparticles by different phenotypes of human macrophages. Flow cytometry is a simple and precise method for quantifying cellular phagocytosis. First, we estimated the effect of different concentrations of bacteria on the phagocytic activity of macrophages. Figure 6A shows histograms of fluorescence distribution for different types of macrophages at various particle–cell ratios. The black line separates the phagocyting cells with high FITC MFI from non-phagocyting cells with low FITC MFI. To set the threshold separating non-phagocyting from phagocyting cells, we performed control experiments with macrophages, which were not incubated with BioParticles (Appendix A). Two parameters were measured for each histogram [27]: (1) the macrophage phagocytic affinity was calculated as the percentage of phagocytic cells; (2) the macrophage phagocytic capacity was estimated as the MFI of phagocytic cells. This parameter is indirectly proportional to the number of engulfed bacterial particles.

Notably, the most significant differences in phagocytic affinity and capacity between groups were observed when bacterial particles were incubated with cells at a ratio of 10:1 (amount of bioparticle per cell) (Figure 6A). An increase in the ratio to 33:1 revealed the decrease in differences in phagocytic affinity, while variations in phagocytic capacity between groups of cells were more intense. When the ratio reached 100:1, only minor differences in MFI were detected between the groups. For further study, we chose the 10:1 ratio because it possessed the greatest differences between the measured parameters.

High variability of phagocytic affinity and capacity among macrophages from different donors did not allow us to perform statistical analysis between the experiments (Appendix A). At the same time, clear differences between groups of macrophages were observed within individual experiments. Thus, we performed a z-standardization of each experiment and used normalized values to analyze phagocytic activity. Figure 6 presents the statistical analysis of phagocytic affinity (B, C) and capacity (D, E) of different macrophage phenotypes. Comparison of both parameters revealed that (1) macrophages differentiated from M-CSF treated monocytes had significantly higher phagocytosis level as compared to GM-CSF treated cells; (2) M-CSF treated cells demonstrated a slight but statistically insignificant decrease in the percentage of phagocytic cells after polarization into M2 phenotype, while the phagocytic capacity remained almost the same for both groups of macrophages; (3) polarization of macrophages into M1 state substantially reduced the level of phagocytosis when compared to M0_GM phenotype. M1 macrophages have the lowest phagocytic activity among all phenotypes studied.

To estimate the impact of growth factors on phagocytic activity, we compared the phagocytosis of M1 macrophages generated after monocyte treatment with GM-CSF and M-CSF (M1 and M1_M, respectively). Analysis of both flow cytometry parameters revealed a significantly higher level of phagocytosis within the M1_M macrophage group compared to the M1_GM phenotype, thereby demonstrating that growth factors exert a significant influence on the functional activity of human macrophages (Appendix A). In addition, we measured TNF-α and IL-6 secretion by M1_M and M1_GM macrophages to investigate the impact of growth factors on the secretory activity of macrophages. The level of both cytokines was significantly higher in the group of M1 macrophages differentiated with M-CSF (Appendix A).

In addition to flow cytometry, fluorescent microscopy was employed to assess the phagocytic activity of macrophages at the 10:1 bioparticles-to-cell ratio. Images of fixed cells obtained with a fluorescent microscope were processed using image analysis software and the percentages of phagocytic cells for different macrophage phenotypes were compared (Figure 7A). The results were similar to the phagocytic affinity (percentage of phagocytic cells) obtained by flow cytometry (compare Figure 6B,C and Figure 7B,C). Notably, a strong positive correlation was observed between the results of flow cytometry and fluorescence microscopy (Figure 7D).

To validate the results obtained with *E. coli* bioparticles, we have measured phagocytic capacity using FITC-conjugated OZ which is non-bacterial phagocytic agent . Fluorescent microscopy revealed that M1 macrophages had lower phagocytic capacity level of OZ particles, comparing to M2 macrophages (Appendix A).

### 2.6. Changes in Mechanical Properties of Macrophages after Activation

Upon activation, macrophages undergo changes in their morphology, which are attributed to alterations in the cytoskeleton and the molecular composition of the cell membrane. The latter changes are due to protein expression, activation of membrane proteins, as well as lipid and protein oxidation, among other factors. The morphological transformation of human macrophages during their activation by PMA is visually seen by both optical microscopy (Appendix A) and AFM (Figure 8).

We compared the following parameters for the different macrophage phenotypes: cell height, stiffness (apparent Young’s modulus), and the viscoelastic parameter power-law exponent (PLE) of the power-law rheology model, which indicates degree of closeness of the behavior of the sample to the liquid (PLE = 1) or solid (PLE = 0).

Firstly, we compared non-activated human macrophages. Various parameters of non-activated human macrophages exhibited variability depending on the cell donor, making it challenging to identify any differences between groups. Nevertheless, significant differences between macrophages were observed only for the cell height parameter. The cell height for both groups of macrophages incubated with M-CSF was significantly lower than that for GM-CSF macrophages, indicating a higher level of flattening (Figure 8 and Appendix A).

Activation of both M1 and M2 macrophages with PMA resulted in a significant increase in cell stiffness (Young’s modulus). The greatest increase of the Young’s modulus, by more than two times from (2.8 ± 1.09) to (6.92 ± 2.87) kPa, was observed for proinflammatory M1 macrophages. Significant changes in the Young’s modulus were also registered in the group of anti-inflammatory M2 macrophages (Figure 9A). However, this parameter did not change significantly after the addition of PMA to precursor macrophages (M0_GM and M0_M). Addition of OZ also led to a significant increase in Young’s modulus in both the M1 and M2 macrophage groups but was insignificant for their precursors (Figure 9B).

Activation of all four macrophage phenotypes with PMA resulted in a reduction of cell height, indicating their flattening (Figure 9). The highest level of flattening was observed for polarized macrophage phenotypes: 1.53- and 1.69-fold decrease of cell height for M1 and M2 macrophages, respectively (Figure 9C). This effect was least pronounced in the M0_GM macrophage group. On the other hand, activation of all macrophage groups with OZ had minimal impact on cell flattening except for a slight decrease observed in the M0_M group (Figure 9D). Particles of the activator can contribute to cell height. With AFM images, it was possible to observe individual particles of OZ on the surface of macrophages.

For all types of macrophages, a decrease in the PLE viscoelastic parameter characterizing the liquid–solid state was registered after activation by PMA (Figure 9E), which means transition to the solid state and an increase in pretension in the cytoskeleton, and usually occurs during cell spreading [44]. The fold of PLE decrease was the lowest for M0_GM cells. Activation of GM-CSF treated macrophages with OZ also led to a decrease in PLE, while M-CSF-treated macrophages demonstrated a slight increase (M0_M) or no change (M2) for this parameter (Figure 9F).

## 3. Discussion

Macrophage activation is involved not only in immune response and tissue regeneration but also in the progression and outcome of many diseases, including autoimmune diseases, cancer, allergic disorders, and different types of infections [12,14]. Therefore, the study of specific macrophage activity is of especially importance [7]. Many studies have been devoted to comparing transcriptomes, proteomes, secreted cytokines, and expressed proteins of different types and phenotypes of macrophages [13,45,46,47,48]. Based on the results, conclusions were drawn about the difference in the functional activity of macrophages. However, macrophage functions are realized through a complex interplay between different proteins, mediators and membrane properties. ROS generation and phagocytosis are the major functions of phagocytes which require a further study.

As expected, functional activities of different types of macrophages differ. Among mouse macrophages, BMDMs were demonstrated to be significantly more phagocytic than other types of macrophages. Zajd et. al. have compared different functional and expression profiles of BMDM and peritoneal murine macrophages (pMAC). Despite some morphological similarities, BMDM have significantly higher phagocytic activity with both *E. coli* Bioparticles and Zymosan particles. ROS-generating activity measured by Amplex Red was almost the same for BMDM and pMAC [49]. A novel real-time imaging platform revealed the higher phagocytosis of BMDM towards different fluorescent pathogen particles as compared to mouse alveolar and peritoneal macrophages [50]. After treatment with fluorescent beads, BMDM displayed an increased rate of phagocytosis and phagosomal proteolysis but a similar oxidative burst compared to RAW 264.7 [51,52].

The majority of macrophage activity studies have been performed using mouse macrophages, while human and mouse macrophages differ significantly in size, immunometabolism, and differentiation stages [48,53,54]. Firstly, human macrophages are primarily derived from blood monocytes, whereas murine studies commonly use macrophages differentiated from bone marrow cells or tissue macrophages [55]. Secondly, markers used for characterizing murine macrophage polarization into M1 and M2 states, such as inducible NOS and arginase, do not serve as reliable markers for polarized human macrophages [7,30]. Lacey et al. (2012) compared the differences between the respective global gene expression profiles of MDM differentiated by GM-CSF or M-CSF with the differences between the respective profiles of murine bone marrow macrophages treated with the same CSF. They found that only 17% of genes regulated differently by these CSFs were common across the species [56].

As for human cells, two types of monocytes are used to obtain macrophages—blood monocytes or THP-1 cells, which is a human monocytic cell line derived from an acute monocytic leukemia patient. THP-1 cells are usually used as a control group for MDM; they are much easier and cheaper to culture, but their responses to activators differ substantially from the response of MDM [57]. Additionally, the markers and functional activity for both types of macrophages are highly dependent on a differentiation protocol [58,59]. While the radical-generating activity of THP-1 macrophages was demonstrated to be 20-fold lower than that of MDM [33], the phagocytic activity was comparable for two types of cells. The percentage of phagocytosis of *E. coli* BioParticles was the same for PMA-differentiated THP-1 cells and MDM [42], and the amount of particles consumed by a cell was significantly lower or comparable for two types of macrophages depending on the cell differentiation protocols and fluorescent particles used [42,43,59]. In our experiments, pro-inflammatory activity of THP-1-derived M1 macrophages was about 100 times lower than that of M1 MDM cells (Appendix A).

Upon activation, macrophages phagocytize foreign material and dead cells and produce reactive oxygen species that contribute, along with enzymes, to the digestion of trapped particles and cells. The primary source of ROS in activated macrophages is NADPH-oxidase which generates superoxide radicals. Concomitant release of SOD3 from intracellular compounds facilitates the dismutation of O_2_^•−^ into hydrogen peroxide. H_2_O_2_ can pass through the cell membrane and diffuse over 100 µm in the extracellular space, affecting endocrine and paracrine redox-dependent signaling [40,60]. H_2_O_2_ turns via Fenton reaction into potent oxidizing agents, like hydroxy and hydroperoxyl radicals (OH● and OOH●), as well as into the corresponding lipid radicals. ROS generated by macrophages can damage not only foreign materials but also host tissues; therefore, one can expect that the ROS generating activity of anti-inflammatory regenerative M2 is significantly lower than the activity of M1 macrophages which produce proinflammatory environment during the acute phase of inflammation.

To compare the level of ROS generation by different macrophage phenotypes, we employed methods of chemiluminescence and fluorescent detection of O_2_^•−^/H_2_O_2_. Advantages of these methods compared to fluorescent intracellular dyes measured by flow cytometry are as follows [20,61]: (1) cell detachment is not required, and (2) the kinetics of radical formation can be continuously monitored. We used substances with different mechanisms of macrophage activation. PMA is a potent activator of protein kinase C, and it is able to penetrate the cell membranes. The high-molecular-weight beta-glucan OZ activates cells through Toll-like type 2 and dectin-1 receptors [62]. Despite the different activation mechanisms, both inducers activated ROS production immediately after addition, and cell responses to both activators were very similar. Significant difference in ROS production and phagocytosis was observed between different macrophage phenotypes.

Exposure of monocytes to GM-CSF or M-CSF leads to different gene expression programs in cells [56], resulting in the appearance of intermediate states of M0 macrophages that are not M1 and M2 phenotypes but to varying degrees have markers and the functional activity of these cells (Figure 1). GM-CSF has low (if any) basal circulating levels that are elevated during immune/inflammatory reactions. GM-CSF is a driver of tissue inflammation, acting on different types of myeloid cells and promoting their differentiation/polarization, such as the maturation of the monocytes into macrophages [2,63]. Incubation of MDM with GM_CSF caused the expression of M1 marker CD86 on the cell surface, but did not activate any of the macrophage functions significantly (Figure 1, Figure 3, Figure 4, Figure 5, Figure 6, Figure 7 and Figure 10). Dormant M0_GM macrophages did not exist for a long time at the inflammatory site. GM-CSF appears almost simultaneously with proinflammatory cytokines and, possibly, LPS. Exposure of M0_GM cells to (+LPS,+IFNꝩ) initiates their polarization into M1 macrophages with the highest expression of the CD86 marker. The function of pro-inflammatory M1 macrophages is to create pro-oxidant environment, they can generate the high level of ROS immediately after activation by any inductor, secrete a high level of proinflammatory cytokine TNF-α but have the lowest ability to perform phagocytosis (Figure 10). With development of inflammation, short-term contact of M0_GM with apoptotic neutrophils can confer features of M2 cells on them [64].

Upon the development of inflammation, the concentration of GM-CSF decreases. M-CSF is constitutively produced by many types of cells including fibroblasts, endothelial cells, monocytes, macrophages, marrow stromal cells, etc., it can be detected in plasma at a concentration of ~10 ng/mL [5]. M-CSF controls the survival, proliferation, differentiation, and functions of monocytes and macrophages [2,65]. In our experiments, M-CSF not only differentiated monocytes but also polarized cells into a M2-like phenotype, which showed different types of M2 activity but did not express CD206 (Figure 1, Figure 3, Figure 4, Figure 5, Figure 6 and Figure 7) [13]. These macrophages possessed the highest phagocytic ability and capacity among all phenotypes. M0_M macrophage treatment with cytokine IL-4 led to an anti-inflammatory macrophage phenotype M2. These cells secreted the highest level of IL-10 and produced radicals in an amount comparable to M1 and M0_M but somewhat more slowly (Appendix A), phagocytized bacteria comparably to M0_M, and expressed CD206 which is a marker of M2 phenotype (Figure 10). Lower ROS production by IL-4 generated M2 phenotype correlated with their highest T cell stimulatory capacity compared to other M2 phenotypes [66].

The effects of cytokines used to polarize macrophages on cell activity have been well documented [7,13,33,67]. M2 macrophages obtained after treatment of M0_M cells with IL-4 or IL-13 differ not only in markers but also in radical-generating activity. Moreover, treatment of M1 with these cytokines alters cell activity to different degrees [68]. Following exposure to proinflammatory cytokines, M0_M macrophages can be polarized into the M1_M phenotype, which express the M1 markers but differs in the secretion of some cytokines [67]. M1_M had much higher phagocytic activity compared to M1 treated with GM-CSF (Appendix A). The results demonstrate that macrophage treatment conditions should be analyzed before comparing the results of different studies.

As demonstrated earlier, activation of macrophages induces changes in their mechanical properties. After exposure to IFN-γ and LPS, an increase in relative elastic modulus was detected in RAW macrophages by optical magnetic twisting cytometry, [38]. An increase (of ~50%) in the Young’s modulus of macrophages adhered to an ECM-coated substrate as compared to an uncoated glass substrate was demonstrated [69]. On the other hand, a decrease in the Young’s modulus of human macrophages was observed upon stimulation with LPS [37]. The results of the latter study, however, can be affected by the high concentration of LPS (10 µg/mL), which induced cytotoxic effects, causing apoptosis and cell softening [70,71]. On the other hand, elasticity of macrophages determining by actin polymerization was demonstrated to affect innate macrophage functions including phagocytosis, secretion of cytokines, and transcriptomic profiles [38].

We analyzed the biomechanical properties of different macrophage phenotypes before and after activation and compared them to the functional activity of the cells. AFM provides a convenient method for the study of the biomechanical properties of the cell surface at the single nanoscale level [36,72]. For non-activated cells, M0_M and M2 macrophages display lower height (greater flattening) compared to M0_GM and M1, respectively. However, the substantial differences in the Young’s modulus and PLE were not observed (Figure 9). When macrophages are activated, they stop migrating, start spreading out, and begin to perform one of their main functions—ROS generation. All macrophage phenotypes revealed a similar decrease in cell height after PMA activation, but there were no changes in cell height after the addition of OZ (Figure 9C,D). Most likely, the particulate agonist with a particle size of 1–3 μm [73], which is present in the medium at high concentrations, can bind to the cell membrane and distort the results of AFM measurements to some degree. The OZ particles attached to the cells can be seen in AFM topography images (Figure 8, bottom row).

Changes in the cytoskeleton and chemical changes in the cell membrane cause changes in the rigidity of macrophages, which are characterized by Young’s modulus. In our experiments, the greatest increase in Young’s modulus was observed for M1 cells. Notably, this phenotype had the highest radical-generating activity. A significant enhancement of stiffness was found in M2-cells. At the same time, Young’s modulus did not change for their counterparts, M0_M and M0_GM, after PMA activation. A decrease in the viscoelastic parameter PLE, which characterizes the liquid–solid transition, was observed after PMA addition to all phenotypes of macrophages (Figure 9E,F). Decrease in PLE means that cell activation initiates pretension in the cytoskeleton, which usually results in macrophage spreading. PLE decrease was also observed for M0_GM and M1 macrophages after OZ addition, while it was insignificant for M0_M and M2.

## 4. Materials and Methods

### 4.1. Cell Culturing

Monocytes were isolated from peripheral blood mononuclear cells (PBMC). Blood was obtained from healthy volunteers. All donors have signed an informed consent form approved by the local ethics committee of Sechenov University (No. 07–17, 13 September 2017, Moscow, Russia).

Briefly, blood was mixed with PBS and layered onto Histopaque-1077 (Sigma Aldrich, Saint Louis, MO, USA) for density gradient centrifugation at 400 g for 40 min. After centrifugation, the PBMCs layer was collected and cells were washed twice with PBS and plated at a density of 7.5–8.5 × 10^5^ PBMCs/cm^2^. Cells were cultured in RPMI complete medium consisting of RPMI-1640 (Corning, Glendale, AZ, USA) supplemented with 10% autologous serum, penicillin (100 U/mL), and streptomycin (100 µg/mL) at 37 °C, 5% CO_2_. After 2 h of incubation, non-adherent cells were washed with PBS, and fresh RPMI complete medium containing 50 ng/mL GM-CSF or 50 ng/mL M-CSF (SCI-store, Moscow, MOW, Russia) was added to adherent monocytes to differentiate them into M0 macrophages (M0_GM and M0_M, respectively). The medium was replaced with the fresh one on the 3rd day of culturing. Before choosing the parameters for macrophage treatment, we reviewed the literature and used the optimal ones to compare four different polar macrophage phenotypes. There is a consensus regarding the concentrations of CSFs and treatment times used for monocyte differentiation into macrophages, but cytokines and LPS used for MDM polarization differ 2–20 times (Appendix A in Appendix A). We applied the middle concentration of cytokines IFN-γ and IL-4 but the lowest concentrations of LPS because LPS was demonstrated to cause dose-dependent apoptosis in macrophages differentiated from THP-1 cells [70]. On the sixth day of culturing, M0 macrophages differentiated with GM-CSF were polarized into M1 macrophages by treatment with 10 ng/mL LPS and IFN-γ 50 ng/mL to the medium, while M0 macrophages differentiated with M-CSF were polarized into M2 macrophages by treatment with IL-4 20 ng/mL [58]. The corresponding precursor M0_GM and M0_M macrophages were kept without any polarization inducers, and only the medium was replaced. After two days of incubation, four phenotypes of MDM were used for the experiments (Figure 11).

### 4.2. Measurement of Surface Markers of Macrophage Polarization

On the 8th day of culturing, MDMs were washed three times with Versene (BioloT, Moscow, Russia) and detached with accutase solution (Sigma Aldrich, Saint Louis, MO, USA) for 5–10 min at 37 °C, 5% CO_2_ to obtain single-cell suspensions and centrifuged at 300× *g* for 10 min. For better cell detachment, cell lifters (SPL Life sciences, Pocheon, Gyeonggi, Republic of Korea) were used. After centrifugation, cell pellets were washed twice with PBS to remove residual accutase, resuspended in flow buffer (PBS + 1% FBS), and aliquoted for subsequent staining with anti-CD206-PE-Cy7 and/or anti-CD86-FITC antibodies (Invitrogen, Waltham, MA, USA) in the dark at 4 °C. After 30 min of incubation, cells were washed in flow buffer and analyzed with Sony SH800 cell sorter (Sony Biotechnology, San Jose, CA, USA). At least 10,000 events were recorded for each sample. Flow cytometry data visualization and analysis were performed with Sony Biotechnology software (version 1.8) and FlowJo (version 10).

### 4.3. Measurement of Secreted Markers of Macrophage Polarization

On the eighth day of culturing, supernatants were collected from MDM and centrifuged at 300× *g* for 10 min to exclude cell debris. ELISA kits for TNF-α and IL-6 determination (“Cytokine”, St. Petersburg, SPB, Russia) were applied to evaluate the concentration of the M1 macrophage markers TNF-α and IL-6 according to the manufacturer’s instructions. To measure the concentration of M2 macrophage marker IL-10, a human IL-10 ELISA kit (Sigma-Aldrich, USA) was used. Optical density was measured using a Multiskan™ FC Microplate Photometer (ThermoFisher, Waltham, MA, USA).

### 4.4. PicoGreen Assay

On the eighth day of culturing, the medium was removed and the attached cells were washed extensively with sterile PBS, the wells were filled with ddH_2_O, and the plate was frozen at −80 °C until measurement. Before the PicoGreen assay, to break down cell membranes and release intracellular DNA, the plates were subjected to three freeze–thaw cycles. Subsequently, the solution from each well was transferred in triplicate to a fresh 96-well plate and the DNA concentration was measured using the Quant-iT PicoGreen dsDNA assay kit (Invitrogen, Waltham, MA, USA). Samples were incubated with PicoGreen for 15 min at room temperature and protected from light. The intensity of DNA fluorescence was measured using a spectrofluorometer Victor Nivo (Perkin Elmer, Shelton, CT, USA) with an excitation wavelength of 480 nm and an emission wavelength of 520 nm.

### 4.5. Measurements of ROS Production by Macrophages

The ability of macrophages to generate ROS was measured using luminol-enhanced CL and Amplex Red, which is converted in peroxidase reaction into fluorescent resorufin. Macrophages were cultured in 24-well plates so that at least three independent measurements were made for each group. The cells were washed twice with PBS to remove the culture medium and debris. After washing, 30 µg/mL horseradish peroxidase and 200 µM luminol were added to each well containing 500 µL of Krebs–Ringer solution with NaHCO_3_ and CaCl_2_ (pH 7.4) for CL measurement, and 60 µg/mL horseradish peroxidase and 10 µM Amplex Red (Sigma Aldrich, Saint Louis, MO, USA) were added for Amplex Red assay. To activate ROS production by macrophages, 300 μg/mL of OZ or 100 ng/mL of PMA were added. We selected concentrations of activators that are usually used to activate phagocytes [41,62,74]. To confirm the selected values, ROS-generating activity was measured at several concentration of activators (Appendix A). The intensity of emitted light, which is proportional to the amount of generated ROS, was recorded immediately after the addition of a macrophage activator by the use of a Victor Nivo multimode plate reader (PerkinElmer, Shelton, CT, USA). Gentle shaking was performed between measurement cycles. To compare the generation of ROS by cells over a certain period of time, the area under the CL or fluorescence curves was measured from the moment the activator was added to the selected time, which exceeded the light peak time. Additionally, we compared ROS generation by detached macrophages measuring CL in cell suspensions by the use of Lum-1200 (DISoft, Moscow, Russia). Macrophages were detached using accutase, resuspended in Krebs–Ringer solution, and transferred to a cuvette at a volume of 480 µL. After that, 400–500 µM luminol and 100–150 µg/mL HRP were added to each cuvette. OZ was added at 300 μg/mL concentration to activate ROS generation. All measurements were performed at 37 °C. For statistical analysis, the areas under the chemiluminescence or fluorescence curves were compared.

### 4.6. Phagocytosis Analysis

Macrophages were cultured in 35 mm dishes on microscopy slides for fluorescence microscopy or in 12-well culture plates for flow cytometry. Phagocytosis was studied using FITC-conjugated *E. coli* bioparticles (Molecular Probes, Eugene, OR, USA) on the eighth day of macrophage cultivation. Bioparticle suspension was thoroughly resuspended and mixed with complete RPMI medium and added to macrophages to achieve a required bacteria-to-cell ratio. Next, the bioparticles were incubated with cells for 1–2 h at 37 °C, 5% CO_2_. After incubation, cells were washed three times with PBS to remove excess bacterial particles. Fluorescence Microscopy: Cells were incubated with ice-cold quenching solution to quench the fluorescence of external BioParticles. Fixation was carried out in 4% paraformaldehyde solution at 4 °C for 20 min. Next, the cells were permeabilized with 0.1% Triton X-100 solution for 5 min at room temperature. To stain the actin cytoskeleton of macrophages, phalloidin–rhodamine was used, and samples were incubated for 2 h in the dark, after which the cells were incubated for 20 min with DAPI to stain the nuclei. After each step, cells were washed three times with PBS. Cells were analyzed using an EVOS M5000 (Thermofisher Scientific, Waltham, MA, USA) fluorescence microscope. Two-class pixel classification with ilastik 1.4.0 was used to create probability maps and CellProfiler 4.2.5 was used to identify macrophages and measure phagocytic affinity. Flow cytometry: Cells were incubated with accutase for 5 min and detached using cell lifters. The obtained cell suspension was centrifuged at 300× *g* for 10 min. Cell pellets were filled with ice-cold quenching solution and incubated for 5 min. After centrifugation, cells were resuspended in fixation buffer (PBS with 0.5% FBS and 1% paraformaldehyde). After 20 min, cells were centrifuged, resuspended in flow buffer (PBS + 1% FBS), and analyzed on a Sony SH800 cell sorter (Sony Biotechnology, USA) using Sony Biotechnology software (version 1.8).

### 4.7. Atomic Force Microscopy

AFM measurements were conducted with a Bioscope Resolve AFM (Bruker, Santa Barbara, CA, USA) mounted on an Axio Observer inverted fluorescent microscope (Carl Zeiss, Oberkochen, Ostalbkreis, Germany). The microscope was equipped with a heated stage, and the sample temperature was kept constant at 37 °C. PeakForce QNM-Live Cell probes (PFQNM-LC-A-CAL, Bruker AFM Probes, Camarillo, CA, USA), short paddle-shaped cantilevers with a pre-calibrated spring constant (average value of 0.1 N/m) were used, the deflection sensitivity (nm/V) was calibrated from the thermal spectrum using the value of the spring constant [75]. The nanomechanical and topography maps were acquired in the fast force volume (FFV) mode with a map size from 20 × 20 to 80 × 80 µm and from 32 × 32 to 128 × 128 point measurements. The force curves (F-Z curves) had a vertical ramp distance of 3 μm, a vertical piezo speed of 183 μm/s, and a trigger force of 0.5–1 nN. 

The numerical processing of the F-Z curves was conducted using Python 3.11.4 scripts (https://github.com/yu-efremov/ViscoIndent accessed on 1 January 2024) developed in the previous works [76,77] with the utilization of the Hertz’s and Ting’s models [78], for the elastic and viscoelastic processing, respectively. The Young’s modulus (E) with the assumptions of the Hertz’s theory, YM (“apparent” elastic modulus), was calculated from the approach part of the force curves:(1)$$F\left(\delta \right)=\frac{4\sqrt{R}}{3(1-\nu^{2})}f_{BEC}\left(\delta \right)E\delta^{\frac{3}{2}}$$
where F is the force acting on the cantilever tip; δ is the indentation depth; ν is the Poisson’s ratio of the sample (assumed to be time-independent and equal to 0.5); R is the radius of the indenter; $f_{BEC}\left(\delta \right)$ is the bottom-effect correction. The same curves were processed with the viscoelastic model:(2)$$F\left(t,\;\delta (t)\right)=\frac{4\sqrt{R}}{3\left(1-\nu^{2}\right)}\int_{0}^{t_{1}\left(t\right)}f_{BEC}\left(\delta \right)E(t-\xi )\frac{d\delta^{\frac{3}{2}}}{d\xi }d\xi$$
(3)$$t_{1}\left(t\right)=\left\{\begin{array}{c} t_{1}\left(t\right)=t,\;0\leq t\leq t_{m}\;\\ \int_{t_{1}\left(t\right)}^{t}E\left(t-\xi \right)\frac{d\delta }{d\xi }d\xi =0,\;t>t_{m} \end{array}\right.$$
where t is the time initiated at the contact; $t_{m}$ is the duration of the approach phase; $t_{1}$ is the auxiliary function determined by Equation (3); ξ is the dummy time variable required for the integration; and $E\left(t\right)$ is the Young’s relaxation modulus for the selected rheology model. Here we used the power law rheology (PLR) model (also known as a springpot in parallel with a dashpot) [79]:(4)$$E\left(t\right)=E_{1}t^{-\alpha }+\eta \delta_{D}\left(t\right),$$
where $E_{1}$ is the relaxation modulus at t = 1 s (scale factor of the relaxation modulus); α is the power law exponent; η is the Newtonian viscous term (with Pa*s units); and $\delta_{D}\left(t\right)$ is the Dirac delta function. A larger α value means a larger amount of relaxation; materials exhibit a solid-like behavior at α=0 and a fluid-like behavior at α=1. The PLR model described by Equation (3) was successfully used for the description of cell mechanics in several previous studies [80].

The topography and local height were calculated from the force curves by the contact point position versus contact position over the substrate, the global tilt correction was performed if needed. We used the top 50% of each cell data set over a cell to define the central part, and the lower areas were discarded from the analysis, since the local properties there were highly affected by the high F-actin concentration at the periphery. From the cell datasets, the mean arithmetic values of YM and $E_{1}$, α, and η were used for further statistical comparison between the samples.

### 4.8. Statistical Analysis

Three technical replicates were measured for each biological sample. R programming language (R version 4.3.1) and GraphPad Prism 9.0.5 were used for statistical data analysis. For experiments on the evaluation of ROS generating capacity, the Mann–Whitney U test was used to compare groups. For phagocytosis experiments, deviations from the expected values were calculated as z-scores. Analysis of variance (ANOVA) with Tukey’s post hoc test and Kruskal–Wallis one-way analysis of variance with Dunn’s post hoc test was applied for multiple comparison. The Shapiro–Wilk test was performed to check if the sample fits the normal distribution. The correlation analysis was conducted using Pearson’s correlation. *p*-values less than 0.05 were considered statistically significant.

## 5. Conclusions

Our study showed that three types of macrophage activity—specific cytokine secretion, ROS generation, and phagocytosis—were not identical for two of the four phenotypes studied. The high variability of the tested parameters evidences that the effects of CSFs and inducers on macrophage polarization differed for different donors. In spite of this variability, we demonstrated a general score of the activities of the macrophage phenotypes. The ability of macrophages to produce ROS can be ranked in the following order: M1 > M2 = M0_M >> M0-GM. The ability of different phenotypes for phagocytosis can be compared as follows: M2 ≥ M0_M > M0_GM > M1. Importantly, the functional activity of macrophage phenotypes is determined not only by polarization inducers but also by growth factors that are used for monocyte differentiation. Notably, M-CSF treated macrophages have significantly higher levels of phagocytosis compared to GM-CSF treated cells (Figure 6, Figure 7 and Appendix A) [27]. Other functions of macrophages, like antigen presentation, protein secretion, and extracellular trap formation, are expected to differ between macrophage phenotypes [81]. Mechanical properties of macrophages are responsible for cell migration and interaction with the extracellular matrix and other cells. Changes in M1 and M2 rigidity indicate another type of intracellular communication compared to their precursors M0_GM and M0_M. Considering the wide variety of different factors that can affect the polarization of macrophages after their differentiation by GM-CSF or M-CSF, such as cytokines, extracellular vesicles, lipid mediators, etc., one can assume a wide variety of different macrophage phenotypes, the functional activity of which is finely tuned to solve specific tasks of different phases of the inflammatory response.

## Figures and Tables

**Figure 1 ijms-25-01860-f001:**
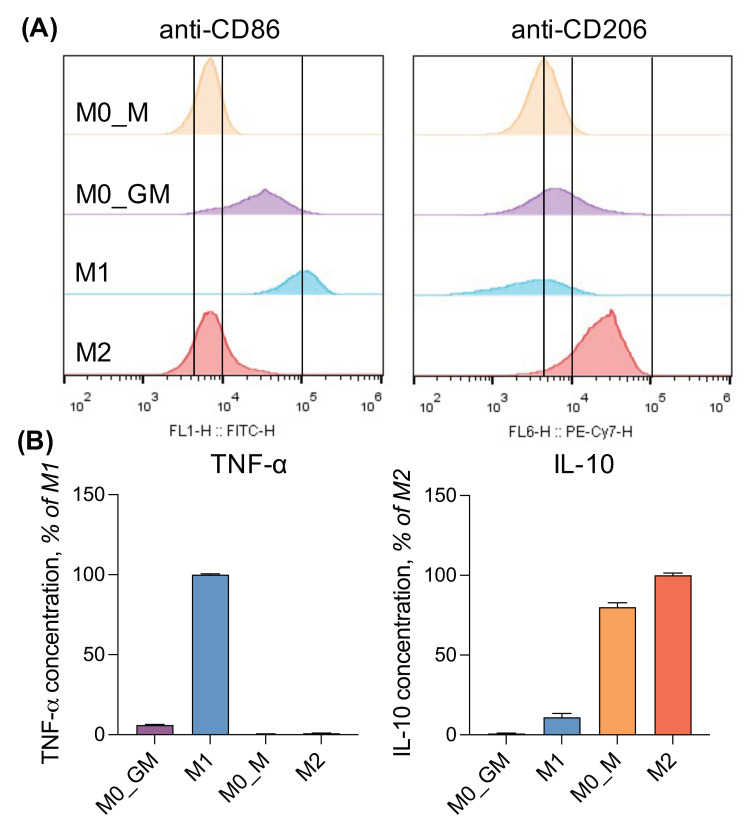
MDM polarization markers. (**A**) Flow cytometry histograms present fluorescence intensity of M0, M1, and M2 macrophages incubated with anti-CD86-FITC or anti-CD206-PE-Cy7 antibodies. Half of the cells cultured with GM-CSF (50 ng/mL) were treated with LPS (10 ng/mL) and IFN-γ (50 ng/mL); cells cultured with M-CSF (50 ng/mL) were treated with IL-4 (20 ng/mL). In untreated cells, the medium was changed (M0_GM and M0_M). In two days of incubation, macrophages were detached by accutase and stained with antibodies. (**B**) Amounts of cytokines TNF-α and IL-10 in conditioning media of macrophages were measured by ELISA.

**Figure 2 ijms-25-01860-f002:**
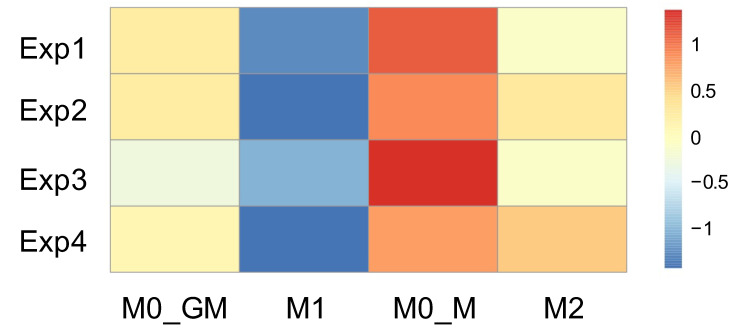
Effect of growth factors and polarizing inducers on the viability of human macrophages. Heatmap presents the amount of double-stranded DNA in a well after cell lysis. The results are presented as z-scores after z-standardization of each experiment.

**Figure 3 ijms-25-01860-f003:**
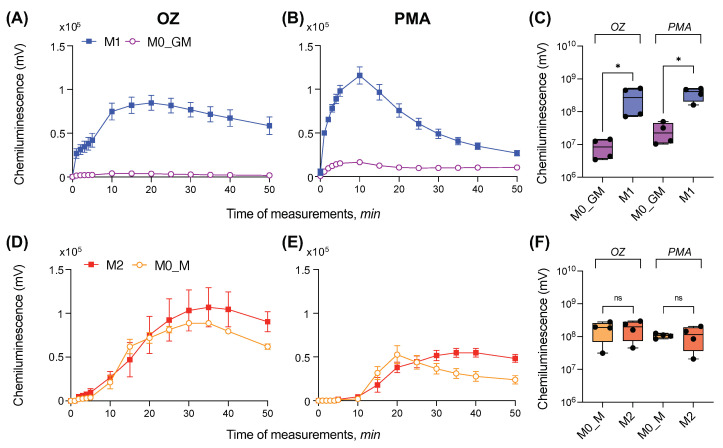
Radical-generating activity of human macrophages measured with luminol-dependent chemiluminescence. (**A**–**C**) Kinetics of CL in one independent experiment (**A**,**B**) and statistical comparison of ROS (**C**) for GM-CSF treated macrophages; (**D**–**F**) kinetics of CL in one independent experiment; (**D**,**E**) and statistical comparison of ROS level (**F**) for M-CSF-treated macrophages. Krebs–Ringer solution with NaHCO_3_, CaCl_2_, and 200 μM luminol was added to each well, and the plate was incubated at 37 °C for 5 min. Measurement of CL was started immediately after the addition of an activator: 0.3 mg/mL OZ (**A**,**D**) or 100 ng/mL PMA (**B**,**E**). Measurements were carried out at 37 °C until the CL maximum was reached for all samples. Each CL curve is the average of three curves obtained by measuring three technical replicates. Y-axis in logarithmic scale. *n* = 4, ^ns^
*p* ≥ 0.05, * *p* < 0.05.

**Figure 4 ijms-25-01860-f004:**
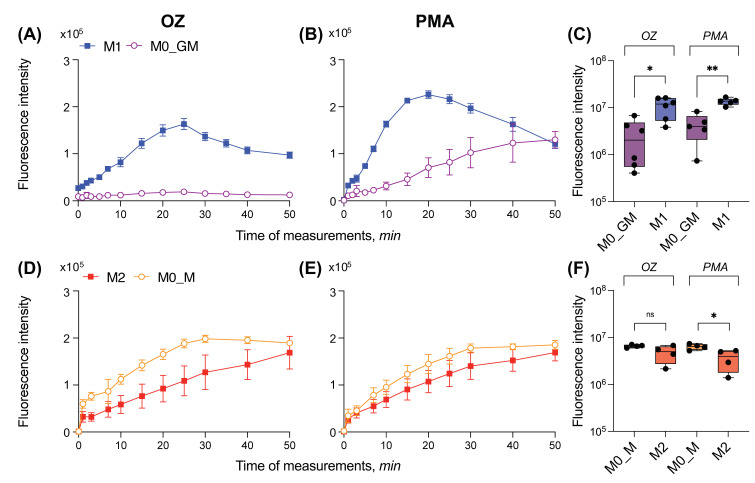
Radical-generated activity of human macrophages measured with Amplex red. (**A**–**C**) Kinetics of Amplex Red oxidation in one independent experiment (**A**,**B**) and statistical comparison of ROS generation (**C**) for GM-CSF treated macrophages; (**D**–**F**) Kinetics of Amplex Red oxidation in one independent experiment (**D**,**E**) and statistical comparison of ROS generation (**F**) for M-CSF treated macrophages. Krebs–Ringer solution with NaHCO_3_, CaCl_2_, and 10 μM of Amplex Red was added to each well, and the plate was incubated at 37 °C for 5 min. Measurement of fluorescence was started immediately after addition of an activator: 0.3 mg/mL OZ (**A**,**D**) or 100 ng/mL PMA (**B**,**E**). Measurements were carried out at 37 °C until the fluorescence maximum was reached for all samples. Each fluorescence curve is the average of three curves obtained by measuring three technical replicates. Y-axis in logarithmic scale. *n* ≥ 4, ^ns^
*p* ≥ 0.05, * *p* < 0.05, ** *p* < 0.01.

**Figure 5 ijms-25-01860-f005:**
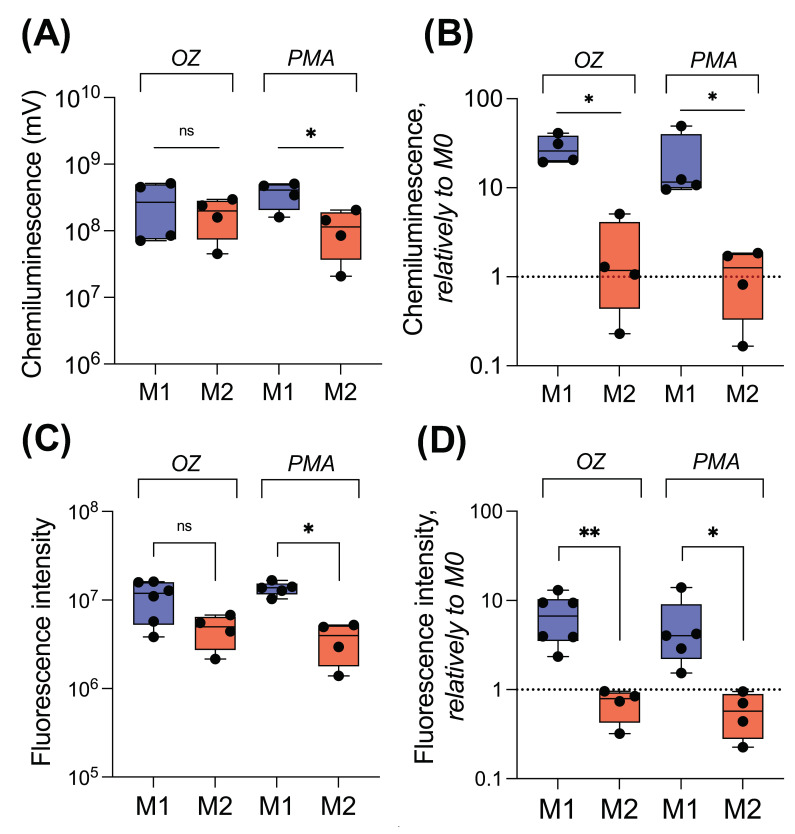
Comparison of radical-generating activity of human macrophages as assessed by CL (**A**,**B**) or Amplex Red oxidation (**C**,**D**). (**A**,**C**) represent the absolute values of the measured parameters; (**C**,**D**) represent the ratio of the measured parameter for M1 or M2 macrophages to the level of that parameter for the corresponding precursor (M0_GM or M0_M, respectively). *n* ≥ 4, ^ns^
*p* ≥ 0.05, * *p* < 0.05, ** *p* < 0.01.

**Figure 6 ijms-25-01860-f006:**
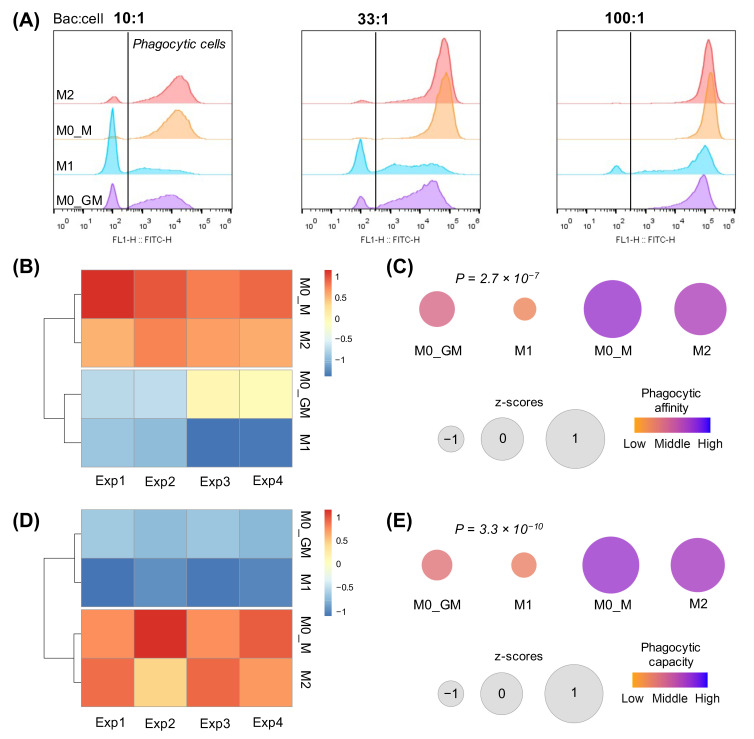
Phagocytosis activity of human macrophages assessed by flow cytometry. (**A**) Histograms of FITC fluorescence intensity distribution after incubation of different groups of macrophages with FITC-labeled E. coli bioparticles. The black vertical line separates the control group of macrophages not incubated with E. coli bioparticles (left) and the group of phagocytic cells (right). The ratio of bacteria to macrophage is indicated above the histograms. (**B**) Heatmap of the percentage of phagocyting macrophages in 4 biological experiments scaled to column z-score. (**C**) Average percentage of phagocyting cells in 4 groups of macrophages. Size and color represent z-score and phagocytic affinity, respectively. M1 versus M0_GM: *p* = 0.00561; M0_M versus M0_GM: *p* < 0.001; M2 versus M0_GM: *p* < 0.001; M0_M versus M1: *p* < 0.001; M2 versus M1: *p* < 0.001; M2 versus M0_M: *p* = 0.303, one-way ANOVA with post hoc Tukey test. (**D**) Heatmap of MFI, which reflects phagocytic capacity, in 4 groups of macrophages scaled to a column z-score. (**E**) Average median fluorescence intensity (MFI) in 4 groups of macrophages. Size and color represent z-score and phagocytic capacity, respectively. M1 versus M0_GM: *p* = 0.0351; M0_M versus M0_GM: *p* < 0.001; M2 versus M0_GM: *p* < 0.001; M0_M versus M1: *p* < 0.001; M2 versus M1: *p* < 0.001; M2 versus M0_M: *p* = 0.4206, one-way ANOVA with post hoc Tukey test).

**Figure 7 ijms-25-01860-f007:**
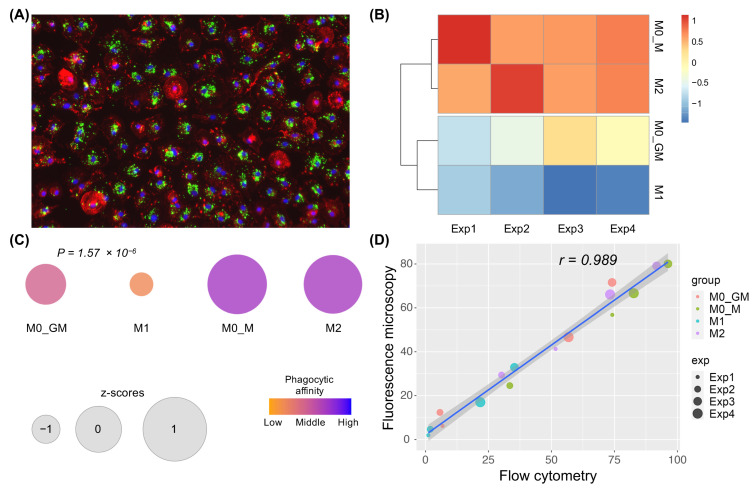
Analysis of phagocytosis activity of human macrophages assessed by fluorescence microscopy. (**A**) Fluorescence microscopy of fixed human macrophages after their incubation with *E. coli* bioparticles at a 10:1 ratio (20×). FITC green—*E. coli* bioparticles; red—phalloidin–rhodamine staining of the actin cytoskeleton; blue—DAPI, nuclear stain. (**B**) Heatmap of the percentage of phagocyting macrophages in 4 groups independent experiments scaled to column z-score. (**C**) Average percent of phagocyting cells in 4 groups of macrophages. Size and color represent z-score and phagocytic affinity, respectively. M1 versus M0_GM: *p* = 0.0038; M0_M versus M0_GM: *p* = 0.0018; M2 versus M0_GM: *p* = 0.0027; M0_M versus M1: *p* < 0.001; M2 versus M1: *p* < 0.001; M2 versus M0_M: *p* = 0.9936, one-way ANOVA with post hoc Tukey test. (**D**) Pearson correlation analysis of phagocytic affinity measured by flow cytometry and fluorescence microscopy. Label “group” is assigned to macrophage group, label “exp” is assigned to one of four biological replicates.

**Figure 8 ijms-25-01860-f008:**
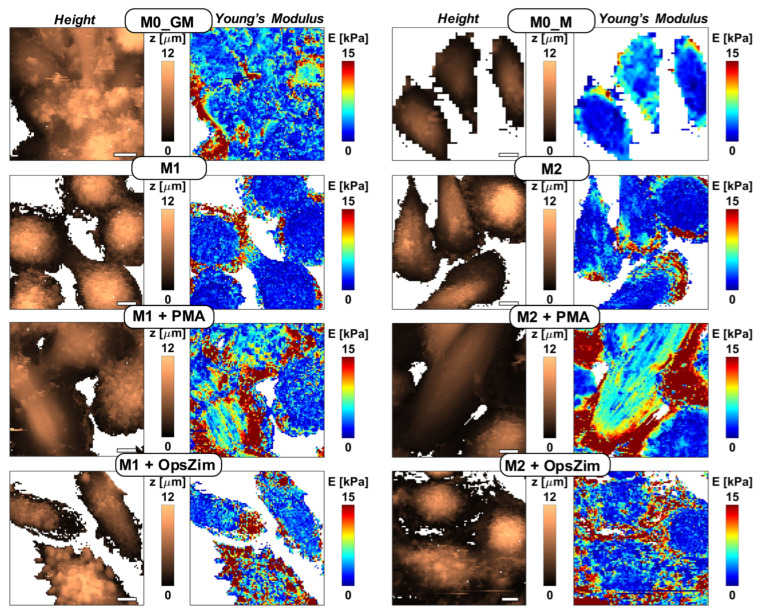
Representative AFM maps obtained on human macrophages untreated and treated with 100 nM PMA or 0.3 mg/mL OZ. Height and nanomechanical maps (distribution of Young’s modulus) of the control and treated cells are presented, z (µm) and E (kPa). Individual particles of OZ on the surface of macrophages can be seen in topography images (bottom row, indicated with arrows). All images have the same color-coded scale for the height and modulus; the scale bar is 10 μm.

**Figure 9 ijms-25-01860-f009:**
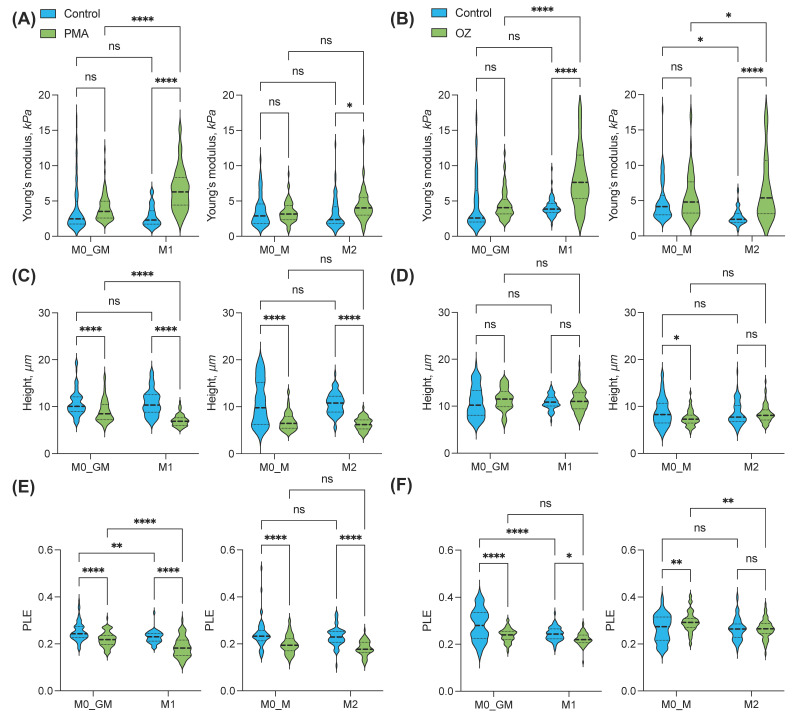
Young’s modulus (**A**,**B**), height (**C**,**D**), and power law exponent (PLE) (**E**,**F**) of human macrophages activated with PMA (left) and OZ (right). Two-way analysis of variance with Tukey post hoc test was performed to compare differences between groups. The analysis includes the results of 2 independent experiments, in each of which the studied parameters were measured for a minimum of 20 individual cells. ^ns^
*p* > 0.05, * *p* < 0.05, ** *p* < 0.01, **** *p* < 0.0001.

**Figure 10 ijms-25-01860-f010:**
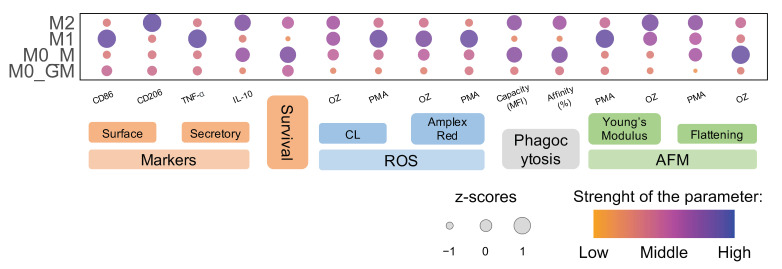
Comparative presentation of markers, functional activity, and mechanical properties of different phenotypes of MDM. Size and color represent z-score for each parameter measured.

**Figure 11 ijms-25-01860-f011:**
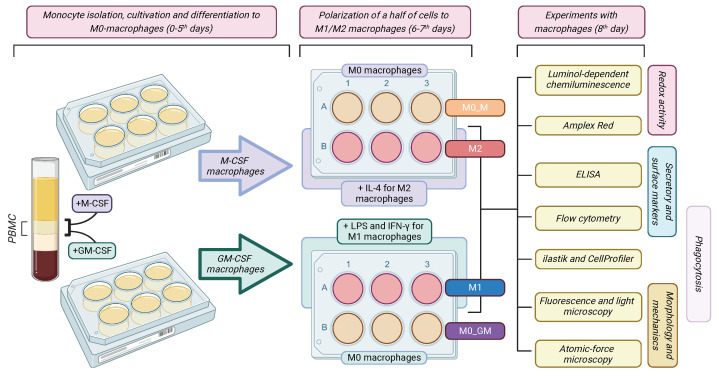
Schematic diagram of the experimental protocol.

## Data Availability

Data generated during the study and included in this article are available from the corresponding authors upon request.

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
