# Peer review of "Radical-Generating Activity, Phagocytosis, and Mechanical Properties of Four Phenotypes of Human Macrophages"

_ijms, 2024, doi:10.3390/ijms25031860_

Round 1

Reviewer 1 Report

Comments and Suggestions for Authors

The authors conducted an extensive investigation into the functional characteristics of M0, M1, and M2 macrophages derived from human PBMCs. However, the ultimate goal of the research remains unclear. Despite various experiments being conducted, the depth of interpretation for each experiment is lacking. Therefore, rigorous additional research and paper revisions are necessary.

-The title suggests the breadth of the study and the ambiguity of the results, necessitating a replacement with a clearer title.

-The absence of control groups and comparison subjects makes the interpretation of the experiment's results challenging. To address this, provide and compare control experiments using mouse bone marrow, RAW, and THP-1.

-While the study utilized a fixed protocol for macrophage differentiation, the possibility of varying reactions due to different concentrations and time points of treated substances was not thoroughly explored. Extensive preliminary research on the concentration and treatment time of substances used in the differentiation process of M0, M1, and M2 macrophages is needed. Additionally, an evaluation of the differentiation protocol used through control experiments is required.

-The phenotypic analysis for M0, M1, and M2 cells lacks sufficient surface protein markers and cytokines. Additional markers are necessary.

-Concentration-dependent experiments, such as PMA and OZ, seem necessary for the study.

-The paper does not present well with repetitive short paragraphs. It is necessary to describe the paper in longer paragraphs, focusing on each topic.

-Minor issues include the unclear conveyance of the use of M0_GM to differentiate M1 and M0_M to differentiate into M2, both in the abstract and the results. Ensure consistency by using "myeloid/monocyte-derived" for MDM. Please check for proper formatting of superscripts, subscripts, italics, etc.

Reviewer 2 Report

Comments and Suggestions for Authors

The authors investigated the polarization of human monocyte-derived macrophages GM-CSF and G-CSF, respectively. GM-CSF-treated macrophages incubated with LPS and IFNg induced polarization toward the M1 phenotype, whereas G-CSF-treated macrophages incubated with IL-4 induced polarization toward the M2 phenotypes. They report that all macrophage populations display distinct phenotypes, including ROS production, cytokine secretion, phagocytosis, and survival. An increase in macrophage stiffness was significant in M1 and M2 stimulated with PMA and opsonized zymosan.

General comment:

Data are informative. Experimenting with human monocyte-derived macrophages is a strength. However, the authors should strengthen the manuscript organization. Adding technical details to the results does not ease reading. The discussion could be more concise by eliminating descriptions of the results already presented in the results section.

Why not use zymosan particles (FITC-labeled zymosan) instead of FITC-conjugated E. Coli bioparticles to monitor phagocytic capacity? Almost all experiments used opsonized zymosan particles. It could make sense to use zymosan particles given the phenotype of M0 macrophages that express high levels of CD206 (mannose receptor).

Specific points:

Line 150: What low molecular PMA is?

Legend Fig. 1; line 178: GM-CSF

Legend Fig. 1; line 179: G-CSF

Line 222: Replace intracellular compounds with intracellular compartments

Lines 226-231: The paragraph is unnecessary. It is a technical point that does not belong to the result section.

Lines 234-235: I suggest to remove this paragraph. The point that luminol monitors intracellular and extracellular ROS (last sentence) is essential.

Legend Fig. 3; line 258: (F) for M-CSF treated macrophages.

Lines 266-269: I would relocate the first and the third sentences in section 4.5.

Lines 275-279: I suggest relocating the paragraph in section 4.5.

Lines 331-335: I am unsure if those sentences are necessary. I would rather be more concise and focus on the main point.

Lines 385-395: This section is misleading. There is no mention of M1_GM and M1_M h before this paragraph. What are the differences between M1_GM, M1_M, M0_GM, and M0-M macrophages? How M1_GM and M1_M macrophages were produced and characterized?

Comments on the Quality of English Language

Minor typos.

Round 2

Reviewer 1 Report

Comments and Suggestions for Authors

There has been a significant improvement in the revision. The authors have thoroughly reviewed my comments and conducted further necessary experiments. No further comment.

Reviewer 2 Report

Comments and Suggestions for Authors

I am satisfied with the changes made to the manuscript and the author's responses.